# SWAN: Sparse Winnowed Attention for Reduced Inference Memory via Decompression-Free KV-Cache Compression

## Abstract

Large Language Models (LLMs) face a significant bottleneck during autoregressive inference due to the massive memory footprint of the Key-Value (KV) cache. Existing compression techniques like token eviction, quantization, or other low-rank methods often risk information loss, have fixed limits, or introduce significant computational overhead from explicit decompression steps. In this work, we introduce SWAN, a novel, fine-tuning-free framework that eliminates this overhead. Our method uses an offline orthogonal matrix to rotate and prune the KV-cache, which is then used directly in the attention computation without any reconstruction. Our extensive experiments demonstrate that SWAN, augmented with a small dense buffer, offers a robust trade-off, maintaining performance close to the uncompressed baseline even at aggressive 50-60% memory savings per-token on KV-cache. A key advantage is its runtime-tunable compression level, allowing operators to dynamically adjust the memory footprint, a flexibility absent in methods requiring fixed offline configurations. This combination of a decompression-free design, high performance under compression, and adaptability makes SWAN a practical and efficient solution for serving LLMs with long contexts.

## 1 Introduction

The remarkable capabilities of Large Language Models (LLMs) are fundamentally linked to their ability to process extensive context lengths (Vaswani et al., 2023; Bai et al., 2024), enabling sophisticated tasks like summarizing long documents. However, this advancement creates a severe inference bottleneck. During autoregressive generation, a Key-Value (KV) cache is essential for storing intermediate attention states to avoid costly recomputation with each new token. For long sequences, the memory required for this cache can vastly exceed that of the model weights, making it the primary performance bottleneck. For instance, a Llama-2 7B model processing a 32k token sequence with a batch size of 16 requires approximately 14 GB for its weights but a staggering 256 GB for its KV cache.

To address this bottleneck, researchers have pursued two main directions. **Architectural methods** such as Grouped-Query Attention (GQA) (Ainslie et al., 2023) improve efficiency but require pre-training. **Post-training approaches** are more flexible but introduce limitations: quantization (Hooper et al., 2025) imposes a fixed compression ceiling, while token eviction (Zhang et al., 2023) risks irreversible information loss, particularly harmful for tasks with long-range dependencies.

A promising direction and the focus of our work leverages a fundamental property of the attention mechanism: the inherent low-rank structure of key and value vectors (Singhania et al., 2024; Saxena et al., 2024). This redundancy in the standard KV-cache opens the door to significant compression. By first using an orthogonal rotation to concentrate the most salient information into fewer dimensions, we can then prune the vector with minimal information loss. Building on this observation, our contributions are as summarized below:

- **A novel, decompression-free framework for KV-cache compression.** We introduce SWAN (Sparse Winnowed Attention), which performs attention directly on a compressed, sparse KV-cache. This eliminates the conventional reconstruction step, removing a significant source of computational overhead present in some prior low-rank methods.

- **A unified approach for simultaneous memory and compute savings.** By using an offline SVD-derived matrix to rotate and prune KV vectors, SWAN's sparse cache can be multiplied directly with dense queries. This design inherently reduces both the memory footprint and the FLOPs required for the attention inner product.

- **Practical runtime adaptability.** SWAN's pruning threshold is a tunable hyperparameter that can be adjusted at inference time, allowing operators to dynamically balance the trade-off between performance and compression without any offline model modification, a crucial flexibility for real-world serving environments.

- **Theoretical and empirical validation.** We provide a theoretical analysis that establishes a clear break-even point for computational savings. We complement this with a detailed empirical analysis on a wide range of NLP benchmarks and model architectures, demonstrating SWAN's effectiveness and robustness in real-world scenarios.

## 2 RELATED WORKS

Efforts to mitigate the KV-cache bottleneck in LLM inference can be broadly categorized into architectural modifications, system-level optimizations, and post-training compression.

**Architectural changes** like Multi-Query and Grouped-Query Attention (MQA/GQA) (Shazeer, 2019; Ainslie et al., 2023) reduce the cache size from the outset but are inapplicable to the vast number of pre-trained models. Orthogonally, system-level solutions like PagedAttention (Kwon et al., 2023) optimize memory management but do not reduce the fundamental size of the cache itself. **Post-training strategies** offer more adaptable solutions. These include Token Eviction methods like StreamingLLM (Xiao et al., 2024) and H2O (Zhang et al., 2023), which discard KV pairs but risk a permanent loss of critical information. Quantization techniques such as KVQuant (Hooper et al., 2025) and KIVI (Zirui Liu et al., 2023) reduce the numerical precision of the cache, but are constrained by a hard upper limit on their compression ratio. **Low-Rank Approximation**, the most relevant area to our work, leverages the insight that KV vectors occupy a low-dimensional subspace. However, existing methods in this space have notable limitations. Approaches like SparQ Attention (Ribar et al., 2024), AQUA Attention (S et al., 2025), and Loki (Singhania et al., 2024) prioritize computational savings over memory reduction. Others, like Eigen Attention (Saxena et al., 2024), tackle the memory issue but require modifying model weights offline for a fixed compression level, sacrificing crucial runtime flexibility. Frameworks such as Lexico (Kim et al., 2024) introduce significant latency by relying on separate compression and decompression steps at every single decoding stage. Our work, SWAN, fills this void with a decompression-free framework that uses a pruned, sparse cache directly in the attention computation. This eliminates reconstruction latency and offers a flexible, runtime-tunable threshold for a balance between performance and compression.

## 3 STANDARD MULTI-HEADED ATTENTION

To establish context for our methodology, we first review the standard self-attention mechanism, which is the core component of Transformer-based LLMs. For a given input sequence of embeddings $X \in \mathbb{R}^{n \times d}$, where $n$ is the sequence length and $d$ is the model dimension, the attention mechanism computes three intermediate representations: the Query ($Q$), Key ($K$), and Value ($V$) matrices. These are derived through linear projections using learned weight matrices $W_Q, W_K, W_V \in \mathbb{R}^{d \times d_h}$, where $d_h$ is the head dimension: $Q = XW_Q, \quad K = XW_K, \quad V = XW_V$. The output of a single attention head is then calculated using the scaled dot-product attention formula (Vaswani et al., 2023): $\text{Attention}(Q, K, V) = \text{softmax}\left(\frac{QK^T}{\sqrt{d_h}}\right) V$.

In practice, Transformers employ a Multi-Head Attention (MHA) mechanism Vaswani et al. (2023), which allows the model to jointly attend to information from different representation subspaces. This is achieved by running the attention mechanism in parallel across $N_h$ independent heads. Each head $j$ has its own set of learned projection matrices $W_Q^{(j)}, W_K^{(j)}, W_V^{(j)} \in \mathbb{R}^{d \times d_h}$. The output of each head is computed as: $\text{head}_j = \text{Attention}(XW_Q^{(j)}, XW_K^{(j)}, XW_V^{(j)})$. The outputs of all heads are then concatenated and projected back to the original model dimension $d$ using an output weight matrix $W_O \in \mathbb{R}^{N_h d_h \times d}$: $\text{MHA}(X) = \text{Concat}(\text{head}_1, \ldots, \text{head}_{N_h})W_O$. The inference process

for an LLM is typically divided into two distinct phases: the prompting (or prefill) phase and the autoregressive decoding phase.

## 3.1 PROMPTING PHASE

In the prompting phase, the model processes the entire input prompt of length $n$ in parallel. For each head, the matrices $Q, K, V \in \mathbb{R}^{n \times d_h}$ are computed for all tokens at once. The attention scores are computed via a large matrix multiplication ($QK^T$), resulting in an attention output for every token in the prompt. This phase is compute-bound due to the large matrix operations. Crucially, at the end of this phase, the computed key and value matrices, $K$ and $V$ for all $N_h$ heads, are stored in memory. This stored data is referred to as the KV-cache.

## 3.2 AUTOREGRESSIVE DECODING PHASE

Following the prompt processing, the model generates output tokens one at a time. For each new token $i + 1$, a single new query vector $q_{i+1} \in \mathbb{R}^{1 \times d_h}$ is computed for each head from the embedding of the previously generated token $i$. Similarly, new key ($k_{i+1}$) and value ($v_{i+1}$) vectors are computed. These new vectors are then appended to the existing KV-cache for each respective head: $K_{\text{cache}}^{(i+1)} = [K_{\text{cache}}^{(i)}; k_{i+1}], \quad V_{\text{cache}}^{(i+1)} = [V_{\text{cache}}^{(i)}; v_{i+1}]$. The attention output for the new token is then calculated using the new query and the entire history stored in the updated cache for each head: $o_{i+1}^{(j)} = \text{softmax}\left(\frac{q_{i+1}^{(j)}(K_{\text{cache}}^{(i+1,j)})^T}{\sqrt{d_h}}\right) V_{\text{cache}}^{(i+1,j)}$. This phase is memory-bandwidth bound. At each generation step, the entire KV-cache, which grows with every new token, must be read from high-bandwidth memory (HBM) into the GPU's faster on-chip SRAM. This data movement constitutes latency and memory bottleneck for long sequences, which our work aims to alleviate.

## 4 METHODOLOGY

Our proposed framework, SWAN as illustrated in Figure 1, operates by rotating the key and value vectors into a low-dimensional subspace where information is maximally concentrated in the initial dimensions. This allows for effective pruning of the later, less important dimensions, leading to significant savings in both memory and computation without a reconstruction step. The core of our method is the offline creation of a powerful projection matrix.

### 4.1 CONSTRUCTION OF PROJECTION MATRICES

The projection matrix is designed to find a basis that aligns the related components of the attention mechanism. Instead of learning separate bases for queries and keys (or values and outputs), we construct a unified basis for their joint distributions. This is a one-time, offline process.

#### 4.1.1 COLLECTING MODEL'S ACTIVATIONS

A representative set of internal activations is collected from the target LLM. A subset of calibration dataset (e.g., BookCorpus) is processed through the model in a single forward pass. For each attention layer $l$, we extract the following: 1. Query ($Q^{(l)}$), and Key ($K^{(l)}$) matrices after the application of Rotary Positional Embeddings (RoPE) (Su et al., 2023), as this reflects their state just before the attention score calculation, 2. Value ($V^{(l)}$) matrices and the output projection weights ($W_O^{(l)}$).

Our method is compatible with both Multi-Head Attention (MHA) and Grouped-Query Attention (GQA). In MHA, the number of query heads ($N_q$) equals the number of key-value heads ($N_{kv}$). In GQA, multiple query heads share a single KV-head ($N_q > N_{kv}$). To create a shared subspace that respects these architectures, we must align the interacting components.

For each KV-head, we group the corresponding $G = N_q/N_{kv}$ query heads. This is achieved by reshaping the query tensor. For a layer $l$, the query tensor $Q^{(l)} \in \mathbb{R}^{N_q \times n \times d_h}$, where $n$ is the number of tokens in the calibration sequence, is reshaped to $Q_{grouped}^{(l)} \in \mathbb{R}^{N_{kv} \times (n \cdot G) \times d_h}$.

A similar alignment is required for the output projection matrix, $W_O^{(l)} \in \mathbb{R}^{N_h d_h \times d}$. This matrix projects the concatenated outputs of all heads back to the model's dimension. Conceptually, we

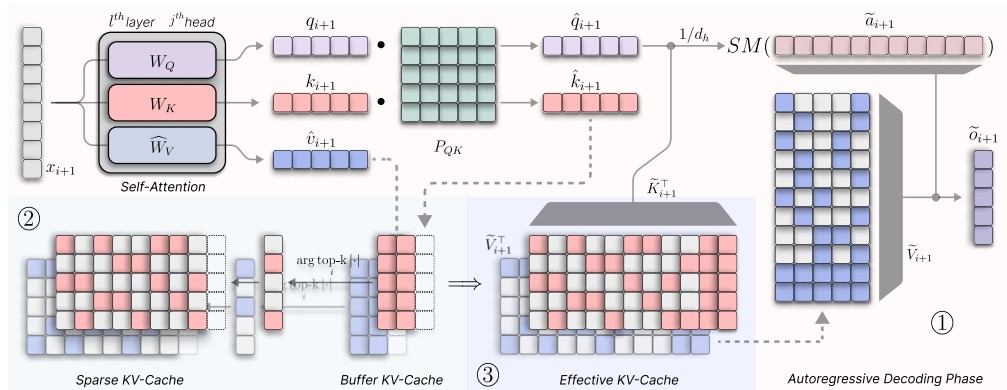

Figure 1: An illustration of the SWAN attention mechanism during a single autoregressive decoding step for token $i+1$. The process begins with the input $x_{i+1}$, where the query ($q_{i+1}$) and key ($k_{i+1}$) are projected at runtime by the orthogonal matrix $P_{QK}$ to produce their rotated counterparts, $\hat{q}_{i+1}$ and $\hat{k}_{i+1}$. The value vector is generated directly in the rotated space as $\hat{v}_{i+1}$ using the pre-modified weight matrix $\widehat{W}_V$. The core of our method is the hybrid KV-cache, composed of two parts: (2) a **Sparse KV-Cache** storing pruned historical vectors, and a small, dense **Buffer KV-Cache** for recent vectors. As new vectors ($\hat{k}_{i+1}, \hat{v}_{i+1}$) enter the buffer, the oldest buffer vector is pruned based on magnitude ('arg top-k') and moved to the sparse cache. The final attention output ($\tilde{o}_{i+1}$) is computed using the rotated query $\hat{q}_{i+1}$ and the (3) **Effective KV-Cache**, which is the combination of both sparse and buffer caches, thus avoiding any decompression overhead.

can slice this matrix into $N_h$ sub-matrices, $\{W_O^{(l,1)}, \ldots, W_O^{(l,N_h)}\}$, where each sub-matrix $W_O^{(l,j)} \in \mathbb{R}^{d_h \times d}$ is responsible for projecting the output of its respective head $j$. To align these components with the value heads for our joint subspace construction, these sub-matrices are grouped in the same manner as the queries, resulting in $N_{kv}$ groups of output weight components, which we denote $W_{O,grouped}^{(l)}$.

### 4.1.2 Forming Joint Subspaces and Deriving the Basis

The central idea is to find a common basis for semantically related tensors. We create two joint matrices for each layer $l$ and each KV-head group $j$: one for Query-Key ($\mathbf{S}_{QK}^{(l,j)}$) and one for Value-Output ($\mathbf{S}_{VO}^{(l,j)}$). These are formed by concatenating the respective (grouped) tensors along the sequence dimension: $\mathbf{S}_{QK}^{(l,j)} = \text{Concat}(Q_{grouped}^{(l,j)}, K^{(l,j)})$, $\mathbf{S}_{VO}^{(l,j)} = \text{Concat}(V^{(l,j)}, W_{O,grouped}^{(l,j)\top})$.

We then apply Singular Value Decomposition (SVD) (Klema & Laub, 1980) to these joint matrices to find their principal components. For a given joint matrix $\mathbf{S}$, SVD provides the decomposition $\mathbf{S} = U\Sigma V^T$. The columns of the right singular matrix $V$ form an orthonormal basis for the row space of $\mathbf{S}$, capturing the directions of greatest variance in descending order. This matrix $V$ becomes our projection matrix: $P_{QK}^{(l,j)} = \text{SVD}(\mathbf{S}_{QK}^{(l,j)})_V$, $P_{VO}^{(l,j)} = \text{SVD}(\mathbf{S}_{VO}^{(l,j)})_V$. These projection matrices, $P_{QK}, P_{VO} \in \mathbb{R}^{d_h \times d_h}$, are computed once offline and stored. During inference, they are used to rotate the attention components into a space where pruning can be performed effectively.

## 4.2 Applying Projection Matrices to Model Weights

To minimize runtime overhead, the projection matrices can be intelligently "absorbed" into the model's original weights before inference begins. However, this optimization is only possible for certain components due to the operational order within the attention mechanism.

The projection for the Value and Output components, $P_{VO}$, can be pre-applied. The original value vector $v$ is produced by $v = xW_V$. The rotated vector is $\hat{v} = vP_{VO}$. We can absorb this rotation into a new weight matrix $\widehat{W}_V$ such that $\hat{v} = x\widehat{W}_V$. This gives $\widehat{W}_V^{(l,j)} = W_V^{(l,j)}P_{VO}^{(l,j)}$. A corresponding transformation is applied to the output projection matrix $W_O$. First, we slice $W_O$ into sub-matrices

---

**Algorithm 1** SWAN Attention (for token i+1)

---

**Require:** Current query $q_{i+1}$, key $k_{i+1}$, value $\hat{v}_{i+1}$ (already projected).
**Require:** KV-Cache: $K_{buffer}, K_{sparse}, V_{buffer}, V_{sparse}$.
**Require:** Projection matrix $P_{QK}$, buffer size $b$, top-k dims $k_{active}$.

1: $\hat{q}_{i+1} \leftarrow q_{i+1}P_{QK}$                        ▷ Project current query
2: $\hat{k}_{i+1} \leftarrow k_{i+1}P_{QK}$                        ▷ Project current key
3: Append $\hat{k}_{i+1}$ to $K_{buffer}$, $\hat{v}_{i+1}$ to $V_{buffer}$.
4: **if** size of $K_{buffer} > b$ **then**
5:      $\hat{k}_{old} \leftarrow$ Pop oldest from $K_{buffer}$.
6:      $\hat{v}_{old} \leftarrow$ Pop oldest from $V_{buffer}$.
7:      $I_k \leftarrow$ arg TopK($|\hat{k}_{old}|, k_{active}$)             ▷ Find top-k indices for key
8:      $k_{sparse} \leftarrow$ Sparse($\hat{k}_{old}, I_k$)               ▷ Create sparse vector
9:      Append $k_{sparse}$ to $K_{sparse}$.
10:     $I_v \leftarrow$ arg TopK($|\hat{v}_{old}|, k_{active}$)            ▷ Find top-k indices for value
11:     $v_{sparse} \leftarrow$ Sparse($\hat{v}_{old}, I_v$)
12:     Append $v_{sparse}$ to $V_{sparse}$.
13: $K_{cache} \leftarrow$ Concat($K_{sparse}, K_{buffer}$)
14: $V_{cache} \leftarrow$ Concat($V_{sparse}, V_{buffer}$)
15: $S \leftarrow \hat{q}_{i+1}K_{cache}^T$            ▷ Sparse-dense & dense-dense mat-vec products
16: $o_{i+1} \leftarrow$ softmax($S/\sqrt{d_h}$)$V_{cache}$         ▷ Compute attention output
17: **return** $o_{i+1}$

---

$\{W_O^{(l,1)}, \ldots, W_O^{(l,N_h)}\}$, one for each of the $N_h$ heads. Since the output of each head will now be in the rotated space ($\hat{o}_j = o_j P_{VO}$), we must modify each weight slice to account for this rotation. The new weight slice for each head $j$ is calculated by pre-multiplying with the transpose of the corresponding projection matrix: $\widehat{W}_O^{(l,j)} = (P_{VO,expanded}^{(l,j)})^T W_O^{(l,j)}$. Here, $P_{VO,expanded}$ is the projection matrix repeated for each query head within a GQA group. These modified sub-matrices are then concatenated back together to form the single new output projection matrix for the layer, $\widehat{W}_O^{(l)}$. This absorption ensures that the value vectors are directly generated in the rotated space and the final projection accounts for this rotation, all with zero computational overhead during inference.

This pre-loading, however, is not feasible for the Query and Key weights. In modern LLMs, Rotary Positional Embeddings (RoPE) Su et al. (2023) are applied to queries and keys *after* their initial projection by $W_Q$ and $W_K$. Since our projection matrices were derived from RoPE-applied vectors, the projection must also occur after RoPE. The sequence of operations is $q \rightarrow \text{RoPE}(q) \rightarrow \text{RoPE}(q)P_{QK}$. Critically, RoPE is a dynamic, position-dependent linear transformation, and matrix multiplication is not commutative. It is not possible to find a static matrix $\widehat{W}_Q$ such that $\text{RoPE}(x\widehat{W}_Q) = \text{RoPE}(xW_Q)P_{QK}$ for all positions. Consequently, the projection $P_{QK}$ must be applied to the query and key vectors at runtime during each decoding step. This introduces a small computational cost. However, this fixed cost is quickly amortized by the substantial computational savings from the subsequent sparse attention calculation. As the sequence length grows, the savings in the dot-product computation, which scale with the sequence length, increasingly outweigh the initial projection cost. We will later provide a theoretical bound for the sequence length at which net computational savings begin. We formally prove that this entire rotation process is lossless in Appendix A.2, demonstrating that the only source of approximation error in our method is the subsequent dimension pruning.

### 4.3 INFERENCE-TIME PRUNING AND ATTENTION COMPUTATION

With the projection matrices established and partially absorbed, our method modifies the standard autoregressive decoding loop. The core innovation is a hybrid KV-cache strategy that combines a large, sparse cache for historical tokens with a small, dense buffer for recent ones. Consistent with findings in prior work (Zirui Liu et al., 2023; Kang et al., 2024; Kim et al., 2024), we observe that maintaining this small buffer of recent tokens in their dense format is crucial for preserving model performance, a finding we empirically validate in our results section.

At each generation step $i + 1$, the new query $q_{i+1}$ and key $k_{i+1}$ are projected into the rotated space at runtime using $P_{QK}$. The new value vector $\hat{v}_{i+1}$ is generated directly in the rotated space, thanks to the absorbed projection in $\widehat{W}_V$. These new, dense projected vectors are temporarily stored in the fixed-size buffer. When the buffer exceeds its capacity, the oldest dense vectors are evicted, pruned, and added to the main sparse cache.

The pruning process is based on magnitude. For an evicted vector (e.g., $\hat{k}_{old}$), we identify the top-$k$ dimensions with the highest absolute values, where $k$ is a tunable hyperparameter controlling the compression ratio. All other dimensions are discarded. The vector is then converted to a sparse representation (e.g., storing only the indices and values of the top-$k$ elements) and appended to the historical sparse cache. This procedure is summarized as Algorithm 1.

This approach yields significant computational savings. The attention score calculation involves multiplying the dense query vector with the hybrid key cache. The operation on the sparse portion of the cache is a sparse-dense matrix-vector product, which is computationally much cheaper than a standard dense-dense operation.

Furthermore, to achieve higher compression rates with minimal performance loss, the non-zero values stored in the sparse vectors can be quantized to 8-bit float. This allows us to retain a larger number of components (a higher $k_{active}$) for the same memory budget. Crucially, unlike token eviction strategies that cause complete information loss for discarded tokens, our method ensures that some information from every token is retained in the cache, preserving a more complete history of the sequence context.

## 5 COMPLEXITY ANALYSIS

### 5.1 SPACE COMPLEXITY

The memory savings of our method are achieved by retaining only the $k_{active}$ most significant components of the historical key and value vectors. We store the explicit values and their corresponding indices for these active dimensions in pre-allocated dense tensors.

Assuming a typical head dimension $d_h = 128$, the indices are stored using 8-bit integers (int8), and the values are stored as 16-bit floating-point numbers (float16). Thus, the memory required for one compressed vector is:

$$M_{sparse} = k_{active} \cdot (\text{sizeof}(\texttt{float16}) + \text{sizeof}(\texttt{int8})) = 3k_{active} \text{ bytes} \tag{1}$$

In contrast, a standard dense vector requires $M_{dense} = d_h \cdot \text{sizeof}(\texttt{float16}) = 128 \cdot 2 = 256$ bytes. The compression ratio is therefore $\frac{3k_{active}}{256}$.

For aggressive compression, the values can be further quantized to 8-bit floats (float8), reducing the memory per vector to $2k_{active}$ bytes. This efficient storage of only specific dimensions allows us to preserve more information per token within the same memory budget compared to dense baselines.

### 5.2 COMPUTATIONAL COMPLEXITY

We theoretically analyze the computational complexity by comparing the number of floating-point operations (FLOPs) required for the attention computation. While our implementation utilizes dense tensor structures for system efficiency, the effective computational complexity is governed by the sparsity of the vectors. Since certain dimensions are zeroed out, they do not contribute to the computation of dot product result. This is an inherent capability of all modern GPU ALUs to efficiently handle these operations, ensuring that zero values are effectively skipped during computation.

Consequently, the complexity break-even point remains consistent with our theoretical model. SWAN becomes efficient than standard attention when the sequence length $L$ satisfies:

$$L > \frac{d_h^2}{d_h - k_{active}} + b \tag{2}$$

Equation 2 defines the trade-off between the fixed projection overhead ($d_h^2$) and the per-token savings ($d_h - k_{active}$). By ignoring zero-value computations, the operational cost scales with $k_{active}$ rather

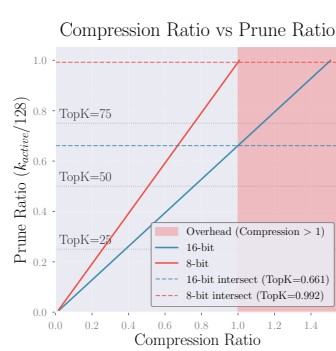
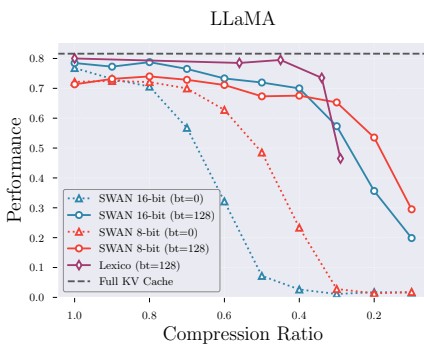

(a) Compression-Pruning tradeoff       (b) Llama-3.1-8B GSM8K results

Figure 2: (a) Relationship between pruning ratio and effective memory compression. (b) Performance of SWAN variants of `Llama-3.1-8B-Instruct` on GSM8K compared to Lexico ($N = 1024$). While Lexico benefits from explicit reconstruction to maintain higher scores at extreme compression, the buffered SWAN variants ('bt=128') demonstrate strong resilience. SWAN maintains competitive performance without the computational overhead of decompression.

than $d_h$. This allows operators to determine the precise context length where SWAN accelerates inference, offering a predictable advantage for long-context applications. The full mathematical derivation is provided in Appendix A.3.

## 6 EXPERIMENTAL RESULTS

### 6.1 THE COMPRESSION-PRUNING TRADE-OFF

In our method, pruning dimensions does not linearly translate to memory savings due to the overhead of storing sparse indices. Figure 2a illustrates this critical relationship. For standard 16-bit values, we must prune over 34% of dimensions (i.e., retain fewer than 66%) just to break even on memory usage. Fortunately, our empirical results (see Appendix A.5) show that SWAN is highly resilient to such pruning, maintaining strong performance even when retaining only half of the original dimensions. This confirms that achieving significant memory savings is practical. Figure 2a also reveals that quantizing values to 8-bit floats makes the compression far more efficient. This insight frames a central question for our experiments: for a given memory budget, is it better to keep fewer, high-precision 16-bit dimensions, or a larger number of less-precise 8-bit ones? The following sections are designed to answer this.

### 6.2 PERFORMANCE ON REASONING TASK

Multi-step reasoning tasks like GSM8K serve as a powerful stress-test for KV-cache compression, as they are exceptionally unforgiving to information loss that can break a model's chain of thought. We use this benchmark to demonstrate SWAN's resilience under pressure on the `Llama-3.1-8B-Instruct` (Grattafiori et al., 2024) model.

We explicitly compare SWAN against **Lexico** (Kim et al., 2024), a state-of-the-art method employing sparse coding with a universal dictionary. For a fair comparison, we evaluate a Lexico variant using a dictionary size of $N = 1024$. The results, presented in Figure 2b, highlight the distinct trade-offs between the two approaches.

Lexico generally exhibits stronger retention of reasoning capabilities at extreme compression ratios. This is structurally inherent to its design: Lexico explicitly *reconstructs* the approximate full-rank KV vectors during attention computation, effectively "filling in the gaps" of the compressed signal to recover lost information. In contrast, SWAN operates under a stricter "what is dropped is lost" constraint. Our method performs attention directly on the sparse elements to eliminate decompression latency, meaning we do not attempt to reconstruct the discarded dimensions. Despite the information loss, SWAN's performance remains close to Lexico's in practical compression regimes

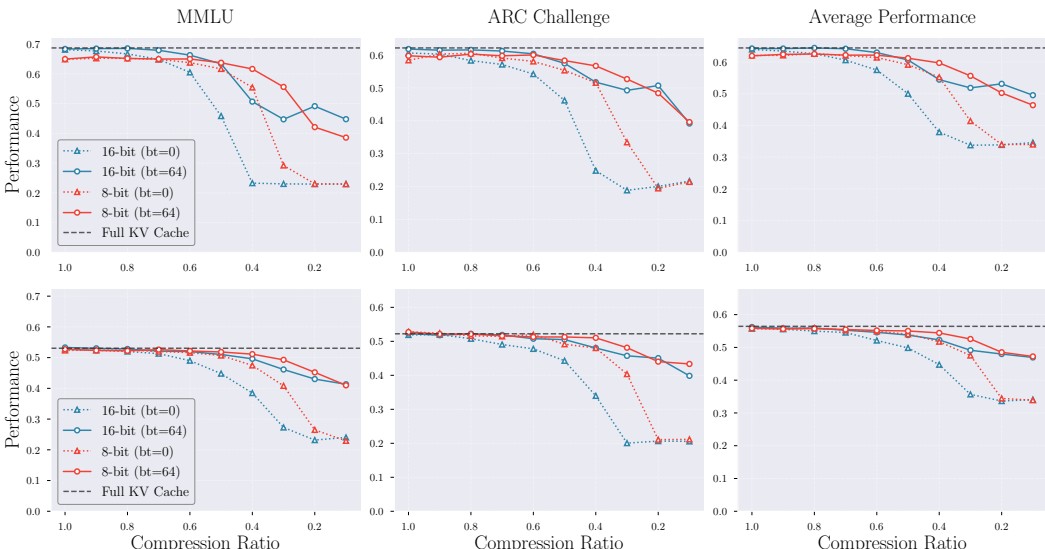

Figure 3: Performance on key NLP benchmarks for `Llama-3.1-8B-Instruct` (top) and `OLMoE-1B-7B-Instruct` (bottom). The buffered SWAN ('bt=64') maintains high performance even at significant compression ratios. Note the consistently smaller performance drop on the sparser OLMoE model, highlighting SWAN's ability to exploit the inherent sparsity in model architectures.

(0.4-0.6 ratio), particularly with the 8-bit buffered variant. SWAN offers a compelling alternative that trades a marginal drop in accuracy for a completely decompression-free inference path.

The figure also highlights the critical role of our hybrid cache design. The zero-buffer variants ('bt=0') suffer a catastrophic performance collapse, confirming that for complex reasoning, a high-fidelity buffer of recent context is non-negotiable. However, with a 128-token buffer, SWAN proves that a decompression-free sparse cache can sustain complex logical chains.

## 6.3 Performance on Standard NLP Benchmarks

To demonstrate SWAN's generalizability, we evaluate it on a range of NLP benchmarks using two distinct architectures: the dense Grouped-Query Attention (GQA) of `Llama-3.1-8B-Instruct` (Grattafiori et al., 2024) and the inherently sparser Multi-Head Attention (MHA) of `OLMoE-1B-7B-Instruct` (Muennighoff et al., 2025). This dual evaluation allows us to assess our method's performance on different attention structures.

The results, shown in Figure 3, reveal three key findings. First, the 64-token dense buffer is critical for performance across all tasks. The buffered variants maintain high accuracy even with 50-60% memory savings, while the zero-buffer versions degrade sharply. Second, for knowledge-intensive tasks like MMLU and ARC Challenge, the 8-bit variant is particularly effective under high compression. Similar to GSM8K task, its ability to retain more, albeit less precise, dimensions proves advantageous for factual recall.

Most importantly, our results reveal a powerful synergy between SWAN and a model's inherent architecture. The performance degradation on the sparser MHA-based OLMoE is consistently lower than on the GQA-based Llama across all tasks. This effect is most dramatically illustrated by the WikiText perplexity benchmark (see Appendix A.8), where the performance drop on OLMoE is three times less severe. This is a crucial finding: SWAN is not just imposing sparsity but is effectively leveraging the natural, learned structure of the model's attention mechanism. It demonstrates that SWAN directly attacks and leverages the *inherent sparsity* of a model's attention mechanism - a capability lacking in previous works. This explains its remarkable performance even without any reconstruction step and makes it an elegant solution for diverse model designs.

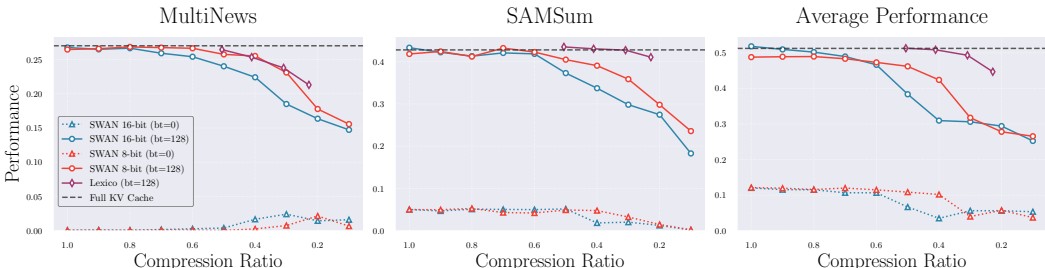

Figure 4: SWAN's performance on the LongBench suite compared to Lexico ($N = 1024$). The figure highlights performance on Multi-News, SAMSum, and the overall Average. While Lexico achieves higher scores due to reconstruction, SWAN 8-bit ('bt=128') remains highly competitive, particularly in summarization tasks. This demonstrates SWAN's ability to capture the 'gist' of long contexts without the computation and latency penalties associated with decompression-based methods like Lexico.

## 6.4 LONG-CONTEXT EVALUATION

Long-context tasks are the ultimate stress-test for a KV-cache compression method, as even minor, cumulative information loss can derail a model's understanding over thousands of tokens. We evaluate SWAN on the LongBench suite (Bai et al., 2024) and compare it against the reconstruction-based Lexico ($N = 1024$) method.

The results in Figure 4 reveal a consistent trend similar to the reasoning benchmarks. Lexico's strategy of reconstructing the cache allows it to maintain a higher performance floor at very aggressive compression ratios (below 0.3). Because Lexico decompresses the cache, it can approximate the original attention scores more closely. However, SWAN demonstrates substantial resilience, particularly in summarization tasks like Multi-News and SAMSum. The 8-bit buffered SWAN variant tracks closely with Lexico's performance until the compression ratio drops significantly. The results imply that for many long-context applications, the computational cost of restoring the full cache yields diminishing returns.

Consistent with previous sections, the 128-token dense buffer ('bt=128') remains critical. The zero-buffer variants ('bt=0') struggle to form a coherent understanding of the initial context, leading to collapse. With the buffer, SWAN proves that a purely sparse, non-reconstructed history is sufficient for high-quality long-context generation.

## 6.5 LATENCY ANALYSIS

Table 1: Latency and Throughput comparison between Eager Baseline and SWAN. SWAN incurs a latency overhead due to the runtime projection and sparse gathering steps, resulting in lower throughput. However, this latency performance cost is balanced against the substantial memory reductions (up to 50-60%) achieved by the method.

| Configuration | Avg Decode Latency (ms) ↓ | Throughput (tokens/s) ↑ |
|---|---|---|
| Standard Attention | 28.41 | 35.11 |
| SWAN ($b = 128, k_a = 64$) | 40.06 | 19.02 |
| SWAN ($b = 64, k_a = 64$) | 40.26 | 18.77 |
| SWAN ($b = 32, k_a = 64$) | 41.09 | 18.37 |
| SWAN ($b = 32, k_a = 32$) | 40.67 | 18.45 |

To evaluate the runtime overhead of our method, we measured the average decoding latency and throughput of SWAN compared to a standard Eager execution baseline. Table 1 presents the results for various configurations of buffer and $k_{active}$ dimensions.

As observed, SWAN introduces an increase in decoding latency (approximately 12ms per token) and a corresponding reduction in throughput. This latency stems primarily from the runtime projection

of queries and keys ($P_{QK}$) and the gathering operations required to manage the sparse cache, which are currently implemented without custom CUDA kernels. Despite this, we argue that this latency is an acceptable trade-off for the massive memory and theoretical compute savings provided by the method, particularly in memory-constrained environments where the baseline would simply OOM. Furthermore, we anticipate that the latency gap can be significantly narrowed through future system-level improvements, such as specialized kernels for sparse-dense matrix multiplication and fused projection-pruning operations.

## 7 CONCLUSION

In this work, we introduced SWAN, a fine-tuning-free framework that fundamentally rethinks KV-cache compression by eliminating the costly, decompression step common to prior low-rank methods. Our approach performs attention directly on a pruned, sparse cache, simultaneously reducing both memory and compute load. Our extensive experiments demonstrated that SWAN, augmented with a small dense buffer, maintains robust performance even at 50-60% memory savings, and uniquely benefits from sparser attention architectures. Its key advantage lies in its runtime-tunable design, which offers a level of operational flexibility absent in methods with fixed offline configurations. As a practical solution for efficient, long-context inference, SWAN's full potential will be further realized with advancements in hardware and software optimized for sparse computation.

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

# A  APPENDIX

## A.1  LANGUAGE MODEL USAGE

We utilized large language models to assist in refining the language and enhancing the clarity of this manuscript, aiming to make our research more accessible and easily digestible for a broad readership.

## A.2  THEORETICAL JUSTIFICATIONS FOR LOSSLESS ROTATION

This appendix provides the formal proofs for the claim that our projection-based rotation of the attention components is lossless prior to the pruning step. We demonstrate that both the attention scores and the final output of the attention mechanism remain unchanged.

**Lemma A.1** (Rotational Invariance of Attention Scores). *Let $P_{QK} \in \mathbb{R}^{d_h \times d_h}$ be an orthogonal projection matrix derived from offline calibration. Let the projected query and key vectors be $\hat{q}_{i+1} = q_{i+1} P_{QK}$ and $\widehat{K}_{cache} = K_{cache} P_{QK}$. The attention scores computed using the original vectors ($S$) are identical to those computed using the projected vectors ($\hat{S}$).*

*Proof.* The original attention scores are given by the dot product $S = q_{i+1} K_{cache}^T$. The attention scores computed with the projected vectors are $\hat{S} = \hat{q}_{i+1} \widehat{K}_{cache}^T$.

We can show that $\hat{S}$ is equivalent to $S$:

$$\begin{aligned}
\hat{S} &= (\hat{q}_{i+1})(\widehat{K}_{cache})^T \\
&= (q_{i+1}P_{QK})(K_{cache}P_{QK})^T && \text{[Substituting definitions]} \\
&= (q_{i+1}P_{QK})(P_{QK}^T K_{cache}^T) && \text{[Using the transpose property } (AB)^T = B^T A^T\text{]} \\
&= q_{i+1}(P_{QK}P_{QK}^T)K_{cache}^T && \text{[Associativity of matrix multiplication]} \\
&= q_{i+1}I K_{cache}^T && \text{[Since } P_{QK} \text{ is orthogonal, } P_{QK}P_{QK}^T = I\text{]} \\
&= q_{i+1}K_{cache}^T = S
\end{aligned}$$

Thus, the scores are identical ($\hat{S} = S$). This proves that the projection of queries and keys is a lossless rotation of the coordinate space that preserves their dot product relationships. $\qquad\square$

**Lemma A.2** (Losslessness of Full Attention with Absorbed Weights). *Let the rotated attention output for a head $j$ be $\hat{o}_j = softmax(S_j/\sqrt{d_h})\hat{V}_j$, where $\hat{V}_j = V_j P_{VO}^{(j)}$. Let the modified output weight be $\widehat{W}_O^{(j)} = (P_{VO}^{(j)})^T W_O^{(j)}$. The final contribution to the MHA output from this head using rotated components is identical to the original.*

*Proof.* The contribution of head $j$ to the final MHA output in the original model is $o_j W_O^{(j)}$. In our modified model, this contribution is $\hat{o}_j \widehat{W}_O^{(j)}$.

From Lemma 1, we know that the attention scores are unchanged by the $P_{QK}$ projection, so the softmax output is also unchanged. Let $A_j = softmax(S_j/\sqrt{d_h})$.

The original head output is $o_j = A_j V_j$. The rotated head output is $\hat{o}_j = A_j \hat{V}_j = A_j(V_j P_{VO}^{(j)})$.

Now, we can show the equivalence of the final output contribution:

$$\begin{aligned}
\hat{o}_j \widehat{W}_O^{(j)} &= (A_j(V_j P_{VO}^{(j)}))(\widehat{W}_O^{(j)}) && \text{[Substituting definition of } \hat{o}_j\text{]} \\
&= (A_j V_j P_{VO}^{(j)})((P_{VO}^{(j)})^T W_O^{(j)}) && \text{[Substituting definition of } \widehat{W}_O^{(j)}\text{]} \\
&= A_j V_j (P_{VO}^{(j)}(P_{VO}^{(j)})^T)W_O^{(j)} && \text{[Associativity of matrix multiplication]} \\
&= A_j V_j I W_O^{(j)} && \text{[Since } P_{VO} \text{ is orthogonal, } PP^T = I\text{]} \\
&= (A_j V_j)W_O^{(j)} \\
&= o_j W_O^{(j)}
\end{aligned}$$

This proves that applying the absorbed weights $\widehat{W}_V$ (to generate $\hat{V}$) and $\widehat{W}_O$ yields the exact same output as the original attention mechanism. Therefore, the only source of approximation error in our method comes from the subsequent pruning of dimensions. $\qquad\square$

### A.3 DERIVATION OF COMPUTATIONAL COMPLEXITY

This appendix provides the detailed mathematical derivation for the computational complexity analysis presented in Section 5.2.

**Proposition A.3** (Complexity of Standard Attention). *The computational cost of the full attention calculation in standard auto-regressive attention for a single head at a sequence length $L$ is $C_{std} \approx 4L \cdot d_h$ FLOPs.*

*Proof.* The total cost consists of two main matrix-vector products and the softmax operation.

1. **Score Calculation:** The product $qK_{cache}^T$ between the query ($1 \times d_h$) and the key cache ($L \times d_h$) costs approximately $2L \cdot d_h$ FLOPs.

2. **Softmax:** The softmax operation has a cost of $O(L)$, which is a lower-order term.

3. **Output Calculation:** The product of the attention weights ($1 \times L$) and the value cache ($L \times d_h$) costs another $2L \cdot d_h$ FLOPs.

Summing these, the total complexity is dominated by the two matrix-vector products, giving $C_{std} \approx 4L \cdot d_h$ FLOPs. □

**Proposition A.4** (Complexity of SWAN). *The computational cost of the full attention calculation in SWAN for a single head at a sequence length $L$ is $C_{SWAN} \approx 4d_h^2 + 4(L-b)k_{active} + 4bd_h$.*

*Proof.* The total cost is the sum of the runtime projections and the two hybrid matrix-vector products for scores and outputs, correctly accounting for sparsity in both the Key and Value caches.

1. **Runtime Projections:** Projecting the current query and key with $P_{QK}$ is a fixed overhead costing $2 \times (2d_h^2) = 4d_h^2$ FLOPs.

2. **Score Calculation:** The computation $S = \hat{q}_{i+1}K_{cache}^T$ involves a sparse-dense product (with the sparse part of the key cache) costing $2(L-b)k_{active}$ FLOPs and a dense-dense product (with the key buffer) costing $2bd_h$ FLOPs.

3. **Softmax:** The cost is again a lower-order term of $O(L)$.

4. **Output Calculation:** The final multiplication of the dense attention scores $(1 \times L)$ with the hybrid value cache $V_{cache}$ has the same structure and cost as the score calculation. It involves a sparse-dense product with the sparse value cache costing $2(L-b)k_v$ (where $k_v = k_{active}$) FLOPs and a dense-dense product with the value buffer costing $2bd_h$ FLOPs.

Summing these costs gives the total complexity:

$$C_{SWAN} \approx 4d_h^2 + [2(L-b)k_{active} + 2bd_h] + [2(L-b)k_{active} + 2bd_h]$$
$$\approx 4d_h^2 + 4(L-b)k_{active} + 4bd_h$$

□

**Proposition A.5** (Computational Break-Even Point). *SWAN is computationally more efficient than standard attention when the sequence length $L$ satisfies: $L > \frac{d_h^2}{d_h - k_{active}} + b$.*

*Proof.* We find the condition for which $C_{SWAN} < C_{std}$, ignoring lower-order terms:

$$4d_h^2 + 4(L-b)k_{active} + 4bd_h < 4Ld_h$$
$$d_h^2 + (L-b)k_{active} + bd_h < Ld_h$$
$$d_h^2 + Lk_{active} - bk_{active} + bd_h < Ld_h$$
$$d_h^2 + b(d_h - k_{active}) < Ld_h - Lk_{active}$$
$$d_h^2 + b(d_h - k_{active}) < L(d_h - k_{active})$$

Assuming $k_{active} < d_h$ (the practical use case), we can divide by the positive term $(d_h - k_{active})$:

$$\frac{d_h^2}{d_h - k_{active}} + b < L$$

This concludes the derivation. □

A.3.1 NUMERICAL EXAMPLES OF THE BREAK-EVEN POINT

To provide a more concrete understanding of the break-even formula, we analyze it under different scenarios. We assume a typical head dimension of $d_h = 128$, which makes the fixed overhead term $d_h^2 = 16,384$.

**Case 1: No Buffer** ($b = 0$) In this scenario, every token past the first is immediately pruned and stored sparsely.

- **Aggressive Pruning (75% pruned, $k_{active} = 32$):** The per-token saving is proportional to $128 - 32 = 96$. The break-even point is $L > \frac{16,384}{96} + 0 \approx 171$ tokens.

- **Moderate Pruning (50% pruned, $k_{active} = 64$):** The per-token saving is proportional to $128 - 64 = 64$. The break-even point is $L > \frac{16,384}{64} + 0 = 256$ tokens.
- **Light Pruning (25% pruned, $k_{active} = 96$):** The per-token saving is proportional to $128 - 96 = 32$. The break-even point is $L > \frac{16,384}{32} + 0 = 512$ tokens.

**Case 2: With Buffer** ($b = 128$) Here, we maintain a dense buffer of the 128 most recent tokens, a common configuration in our experiments. The buffer size is simply added to the break-even point calculated from the pruning ratio.

- **Aggressive Pruning (75% pruned, $k_{active} = 32$):** The break-even point is $L > 171 + 128 = 299$ tokens.
- **Moderate Pruning (50% pruned, $k_{active} = 64$):** The break-even point is $L > 256 + 128 = 384$ tokens.
- **Light Pruning (25% pruned, $k_{active} = 96$):** The break-even point is $L > 512 + 128 = 640$ tokens.

These examples clearly illustrate the trade-off: more aggressive pruning leads to substantial per-token savings, allowing the system to overcome the fixed projection overhead much earlier in the sequence. The dense buffer adds a constant offset to this point, representing the initial context length during which all attention computations remain dense.

Table 2: Performance of `Llama-3.1-8B-Instruct` as a function of the top-k retention ratio ($k_{active}/d_h$). A ratio of 1.0 (B) is the uncompressed baseline. The best performance for each task is highlighted in bold. Acronyms: HS (HellaSwag), WN (Winogrande), TQA (TruthfulQA), ARC-C (ARC Challenge), WT (WikiText). Arrows indicate whether a higher ($\uparrow$) or lower ($\downarrow$) score is better.

| Ratio | MMLU $\uparrow$ | GSM8K $\uparrow$ | HS $\uparrow$ | WN $\uparrow$ | TQA $\uparrow$ | ARC-C $\uparrow$ | WT $\downarrow$ | Avg Perf. $\uparrow$ |
|---|---|---|---|---|---|---|---|---|
| 1.0 (B) | **0.687** | **0.804** | **0.608** | **0.762** | **0.551** | **0.621** | **8.912** | **0.671** |
| 0.9 | 0.687 | 0.781 | 0.607 | 0.757 | 0.550 | 0.617 | 8.913 | 0.666 |
| 0.75 | 0.683 | 0.792 | 0.606 | 0.762 | 0.551 | 0.611 | 8.943 | 0.668 |
| 0.5 | 0.659 | 0.642 | 0.596 | 0.735 | 0.536 | 0.583 | 9.462 | 0.625 |
| 0.3 | 0.310 | 0.038 | 0.456 | 0.545 | 0.491 | 0.391 | 21.097 | 0.372 |

### A.4 BENCHMARK DETAILS

This section provides details on the evaluation benchmarks used throughout our experiments, including the number of few-shot examples and the primary evaluation metric for each task.

MULTI-TASK BENCHMARKS

- **MMLU (Massive Multitask Language Understanding)** (Hendrycks et al., 2021): A diverse benchmark testing world knowledge and problem-solving skills across 57 subjects. Evaluated with 5-shot prompting and measured by accuracy.
- **HellaSwag** (Zellers et al., 2019): A commonsense reasoning task that involves choosing the most plausible continuation of a sentence. Evaluated with 10-shot prompting and measured by accuracy.
- **Winogrande** (Sakaguchi et al., 2019): A benchmark designed to test commonsense reasoning through pronoun resolution problems. Evaluated with 5-shot prompting and measured by accuracy.
- **TruthfulQA (MC2)** (Lin et al., 2022): A task that measures a model's tendency to answer questions truthfully, even when a common misconception provides a tempting alternative. Evaluated with 6-shot prompting and measured by accuracy.
- **ARC Challenge** (Clark et al., 2018): A question-answering dataset composed of challenging science questions from grade-school to high-school level. Evaluated with 25-shot prompting and measured by accuracy.
- **WikiText** (Merity et al., 2016): A benchmark measuring language modeling quality through zero-shot evaluation, measured by word-level perplexity, where lower is better.

REASONING BENCHMARK

- **GSM8K** (Grade School Math 8K) (Cobbe et al., 2021): A dataset of grade-school math word problems that require multi-step reasoning. Evaluated with 5-shot prompting and measured by the flexible extract version of exact match of the final answer.

LONG-CONTEXT BENCHMARKS (LONGBENCH v1 (BAI ET AL., 2024))

- **Multi-News**: A long-document summarization task. Performance is measured by ROUGE score (Lin, 2004).
- **TREC**: A fine-grained question classification task on long documents. Performance is measured by classification score.
- **LCC** (Code Completion): A task evaluating a model's ability to complete long code snippets. Performance is measured by code similarity score.
- **Passage Retrieval (en)**: A task that requires retrieving relevant passages from a long document. Performance is measured by retrieval score.
- **SAMsum**: A dialogue summarization task. Performance is measured by ROUGE score.

## A.5 PERFORMANCE VS. PRUNING RATIO ANALYSIS

To establish the trade-off between KV-cache compression and model performance, we evaluate the `Llama-3.1-8B-Instruct` (Grattafiori et al., 2024) model's performance degradation as we vary the percentage of dimensions retained ($k_{active}$) in the sparse cache. The results, shown in Table 2, benchmark our method across several common-sense reasoning, knowledge, and language modeling tasks. A retention ratio of 1.0 represents the uncompressed baseline performance.

The results demonstrate that performance remains remarkably stable with moderate pruning. Retaining 75% of the dimensions ('Ratio=0.75') results in a negligible drop in average performance, staying within 1% of the baseline. Even when pruning half of the dimensions ('Ratio=0.5'), the model retains strong performance across most tasks, with an average degradation of less than 5%.

However, performance begins to degrade more significantly with more aggressive pruning. At a retention ratio of 0.3 (a 70% reduction in dimensions), we observe a sharp decline across all benchmarks. Notably, the GSM8K task, which evaluates mathematical reasoning, is the most sensitive to information loss. Its performance collapses almost completely at the 0.3 ratio, dropping from over 80% to just 3.8% accuracy. This high sensitivity makes GSM8K an excellent benchmark for stress-testing our method. Consequently, our analysis in the main paper begins on this task, as its performance can be considered a practical lower bound for our approach's capabilities.

## A.6 ABLATION STUDY ON KEY AND VALUE PRUNING RATIOS

To understand the relative importance of the key and value vectors in preserving model performance, we conducted an ablation study where we varied the proportion of dimensions retained for each. The study was performed using the `meta-llama/Llama-3.1-8B-Instruct` model with a dense buffer size of zero ($b = 0$) to isolate the effect of the sparse cache. We evaluated the model on a suite of standard benchmarks, each with a specific few-shot setting to ensure fair comparison.

The results of this study are presented in Table 3. We vary the retention ratio for keys ($TopK_R$) and values ($TopV_R$) such that their sum is always 1.0, representing a fixed information budget distributed between them.

The results clearly indicate that both key and value vectors are critical for retaining model performance. Extreme pruning of either component leads to a dramatic drop in accuracy across all tasks. For instance, retaining only 10% of key dimensions ($TopK_R = 0.1$) while keeping 90% of value dimensions results in a catastrophic increase in WikiText perplexity and poor performance on all other benchmarks.

Interestingly, the optimal balance appears to be near the center. The configuration with symmetric pruning ($TopK_R = 0.5, TopV_R = 0.5$) achieves the best or near-best results on every single task, suggesting that, as a general rule, keys and values are of roughly equal importance. The slightly asymmetric configuration of ($TopK_R = 0.6, TopV_R = 0.4$) also performs exceptionally well,

Table 3: Ablation study on the `meta-llama/Llama-3.1-8B-Instruct` model with a zero-token buffer ($b = 0$), varying the retention ratios for Key ($TopK_R$) and Value ($TopV_R$) vectors. The best performance for each task is highlighted in bold. Acronyms: MMLU (Massive Multitask Language Understanding), HS (HellaSwag), WN (Winogrande), TQA (TruthfulQA MC2), ARC-C (ARC Challenge), and WT (WikiText). Arrows indicate whether a higher ($\uparrow$) or lower ($\downarrow$) score is better.

| $TopK_R$ | $TopV_R$ | MMLU $\uparrow$ | HS $\uparrow$ | WN $\uparrow$ | TQA $\uparrow$ | ARC-C $\uparrow$ | WT $\downarrow$ |
|---|---|---|---|---|---|---|---|
| 0.1 | 0.9 | 0.23 | 0.26 | 0.50 | 0.48 | 0.21 | 1431.04 |
| 0.2 | 0.8 | 0.23 | 0.30 | 0.53 | 0.47 | 0.24 | 47.09 |
| 0.3 | 0.7 | 0.50 | 0.55 | 0.59 | 0.47 | 0.49 | 13.01 |
| 0.4 | 0.6 | 0.63 | 0.59 | 0.68 | 0.52 | 0.57 | 10.02 |
| 0.5 | 0.5 | **0.66** | **0.60** | **0.73** | **0.54** | **0.58** | **9.46** |
| 0.6 | 0.4 | **0.66** | 0.59 | **0.73** | 0.52 | **0.58** | 9.52 |
| 0.7 | 0.3 | 0.64 | 0.58 | 0.72 | 0.50 | 0.56 | 10.11 |
| 0.8 | 0.2 | 0.57 | 0.52 | 0.66 | 0.50 | 0.49 | 12.48 |
| 0.9 | 0.1 | 0.24 | 0.32 | 0.52 | 0.48 | 0.26 | 63.30 |

matching the top performance on MMLU, Winogrande, and ARC-C. This might suggest that for some tasks, retaining slightly more information in the key vectors, which are responsible for the attention score distribution, is marginally more beneficial than retaining it in the value vectors, which carry the content to be aggregated. However, the overall trend strongly supports a balanced pruning strategy as a robust and effective baseline for our method.

## A.7 ABLATION STUDY: IMPORTANCE OF PROJECTION MATRIX SPECIFICITY

While any orthogonal projection is mathematically lossless before pruning, the performance of our method hinges on the quality of the rotation itself. A well-chosen rotation aligns the most important information with the top dimensions, minimizing information loss during the subsequent pruning step. To validate that our data-driven method for computing projection matrices is critical for performance, we conduct a series of ablation studies. We compare our proposed projection method against several variants, all evaluated with a 50% retention ratio ($k_{active}/d_h = 0.5$).

The ablation experiments are designed as follows:

- **Random Projection:** We replace our learned matrices with an orthogonal matrix derived from 4096 randomly generated vectors following a Gaussian distribution. This tests whether a generic, non-data-driven orthogonal basis is sufficient.

- **Layer-Shuffle:** We randomly shuffle our pre-computed projection matrices across the model's layers. This experiment is designed to determine if the learned subspaces are specific to each layer or if a more generic, layer-agnostic projection would suffice.

- **KV-Shuffle:** We interchange the projection matrices for the Key-Value and Query-Output subspaces ($P_{QK} \leftrightarrow P_{VO}$). This tests whether the learned subspaces for these distinct components are interchangeable or highly specialized.

- **Head-Shuffle:** Within each layer, we randomly shuffle the projection matrices among the different attention heads. This evaluates if the learned subspaces are specific to individual heads or are generalizable across all heads in a layer.

The results in Table 4 unequivocally demonstrate the superiority of our tailored projection method. Our approach outperforms all ablation variants across every benchmark, confirming that the specificity of the projection is crucial for retaining model performance.

**Key Findings:**

- The **Random Projection** yields the most significant performance degradation, proving that the projection matrix must be data-driven and derived from the model's actual activation patterns. A generic orthogonal basis is insufficient for identifying and preserving salient information.

Table 4: Ablation results comparing our proposed projection method against several variants at a 50% retention ratio. Our proposed method consistently outperforms all others, highlighting the importance of its data-driven and component-specific nature. Acronyms: HS (HellaSwag), WN (Winogrande), TQA (TruthfulQA), ARC-C (ARC Challenge), WT (WikiText).

| Projection Method | MMLU ↑ | HS ↑ | WN ↑ | TQA ↑ | ARC-C ↑ | WT ↓ | Avg Perf. ↑ |
|---|---|---|---|---|---|---|---|
| **Our Projection** | **0.66** | **0.60** | **0.73** | **0.54** | **0.58** | **9.46** | **0.62** |
| Head-Shuffle | 0.60 | 0.57 | 0.70 | 0.50 | 0.56 | 10.72 | 0.59 |
| Layer-Shuffle | 0.59 | 0.57 | 0.68 | 0.51 | 0.56 | 10.85 | 0.58 |
| KV-Shuffle | 0.58 | 0.57 | 0.69 | 0.49 | 0.54 | 11.24 | 0.57 |
| Random Projection | 0.57 | 0.57 | 0.68 | 0.49 | 0.54 | 11.13 | 0.57 |

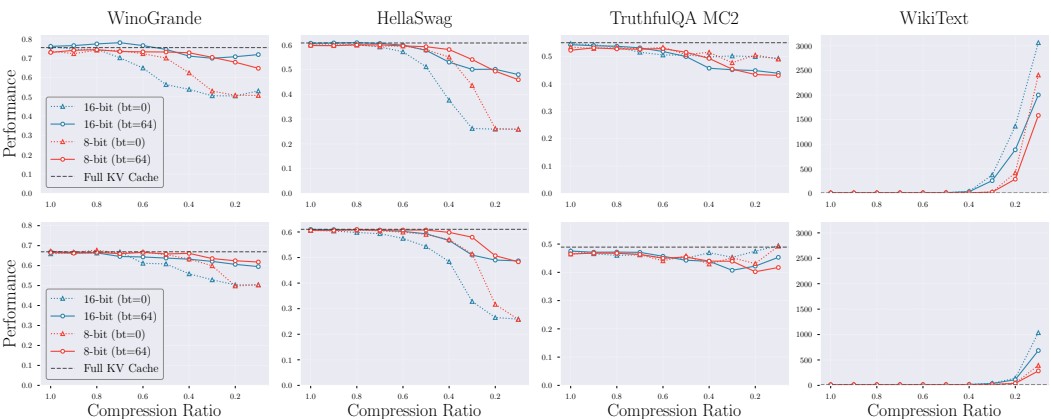

Figure 5: Detailed performance on additional NLP tasks for `Llama-3.1-8B-Instruct` (top row) and `OLMoE-1B-7B-0924-Instruct` (bottom row). The figure displays results for Winograde, HellaSwag, TruthfulQA MC2, and WikiText. The trends confirm the critical role of the dense buffer in preserving performance across different task types and model architectures.

- The performance drops from **shuffling across layers, heads, and KV-components** confirm that the learned low-dimensional subspaces are highly specialized. This validates our approach of creating distinct projection matrices for each specific component (per layer, per head, and for QK/VO subspaces separately), as these structures are not interchangeable.
- Even the seemingly minor drop in the **Head-Shuffle** experiment highlights the fine-grained nature of attention. Each head learns to focus on different aspects of the input, and our method successfully captures this specialized structure.

In conclusion, this ablation study validates that the performance of our method is not merely a consequence of using an orthogonal transformation but is critically dependent on our careful, data-driven process for constructing matrices that are specific to each component of the attention mechanism.

### A.8 DETAILED RESULTS ON ADDITIONAL NLP BENCHMARKS

This section provides the detailed performance plots for the NLP benchmarks not included in the main body of the paper, evaluated on both `Llama-3.1-8B-Instruct` (top row) and `OLMoE-1B-7B-0924-Instruct` (bottom row). The results on commonsense reasoning (Winograde, HellaSwag), model truthfulness (TruthfulQA), and language modeling (WikiText) are fully consistent with the findings discussed in Section 6.3.

The detailed results in Figure 5 not only reinforce our main findings but also reveal fascinating, nuanced interactions between SWAN, model architecture, and task-specific requirements. A key insight emerges from comparing the Multi-Head Attention (MHA) of OLMoE with the Grouped-Query Attention (GQA) of Llama. OLMoE's architecture, with unique Key and Value projections for each head, is inherently more sparse than Llama's, where dense Key and Value vectors are shared across groups of queries. As SWAN is designed to directly exploit sparsity, it naturally exhibits a

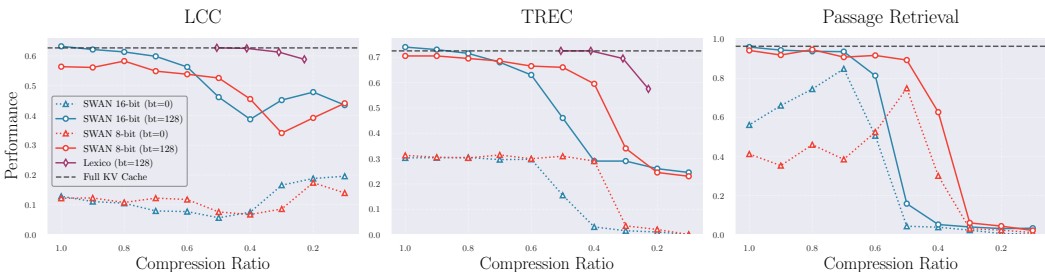

Figure 6: Performance on additional LongBench tasks. The buffered SWAN ('bt=128') confirms its effectiveness on code completion (LCC), classification (TREC), and passage retrieval, demonstrating the generalizability of our approach for diverse long-context scenarios.

more graceful performance degradation on the sparser OLMoE architecture, a compelling trend observed across all tasks.

On the commonsense reasoning benchmarks, SWAN demonstrates exceptional resilience. For **Winogrande**, performance remains remarkably stable, showing very little drop even at the highest compression levels. Even the zero-buffer versions maintain their integrity until a 60% compression ratio, highlighting the task's robustness to information pruning. For **HellaSwag**, the buffered variants show almost no performance loss down to a 40% compression ratio, after which a clear threshold is crossed and performance drops sharply. This suggests a critical point of information density required for this specific reasoning task.

The **TruthfulQA MC2** benchmark presents a unique case. Both buffered and non-buffered variants are surprisingly stable, with only a minor, consistent drop after a 50% compression ratio. This indicates that the information required for factual consistency is highly concentrated within the most energetic components of the KV vectors and is less dependent on the high-fidelity recent context provided by the buffer.

Finally, the highly sensitive **WikiText** perplexity benchmark serves as a powerful testament to our method's architectural advantages. While performance holds steady until a 40% compression ratio for Llama and 30% for OLMoE, the subsequent degradation is dramatically different. The perplexity spike on OLMoE is three times less severe than on Llama. This is a crucial piece of evidence supporting our central claim: SWAN's design, which thrives on sparsity, is inherently more efficient and less disruptive on models with sparser attention mechanisms like MHA.

## A.9 DETAILED LONGBENCH TASK RESULTS

This section provides detailed results for the remaining LongBench tasks, which confirm the conclusions drawn in the main paper. As shown in Figure 6, the trends are consistent across code completion (LCC), fine-grained classification (TREC), and passage retrieval.

Across all tasks, the 128-token buffer is essential for avoiding performance collapse, and the buffered SWAN variants exhibit a robust and graceful trade-off between compression and accuracy. The specific degradation patterns vary by task, suggesting different sensitivities to information loss. For instance, performance on the TREC classification task shows a particularly sharp drop after 50% compression, indicating a high reliance on specific details that are pruned more aggressively at that point. Nonetheless, the overall effectiveness of our hybrid-cache approach is validated across these diverse, long-context challenges.

