# OpenReview forum: "SWAN: Sparse Winnowed Attention for Reduced Inference Memory via Decompression-Free KV-Cache Compression"
_ICLR.cc/2026/Conference — Submitted to ICLR 2026_

### Official Review · Reviewer_rx4F · 2025-10-30

**Soundness:** 3
**Presentation:** 3
**Contribution:** 3
**Rating:** 4
**Confidence:** 5

**Summary:**

This paper proposes to improve memory footprint of the KV-cache, by using an offline orthogonal matrix to rotate and prune the KV-cache. SWAN (Sparse Winnowed Attention) is introduced to perform attention directly on a compressed, sparse KV-cache without (conventional) reconstruction, called "decompression-free" as indicated in the paper title. Being free of decompression/reconstruction leads to simultaneous memory and compute savings -- this claim and contribution sound promising!

**Strengths:**

1. SWAN can achieve up to 50-60% memory savings due to the improvement in memory footprint of KV-cache. Being free of decompression/reconstruction leads to simultaneous memory and compute savings. In other words, unlike other existing decomposition-based KV-cache compression methods, SWAN incurs no (or minimal) computation overhead for reconstruction, and thus save compute as well.

**Weaknesses:**

1. Being free of decompression/reconstruction leads to simultaneous memory and compute savings -- this claim and contribution sound promising! However, if the authors claim to save compute, then they should also demonstrate results of FLOPs (# of floating-point operations), LLM runtime (end-to-end latency), and speed-up (percentage of latency improvement); but these are not thoroughly discussed, nor experimentally analyzed in the paper.
2. Direct comparisons (with experiments/results) against other related works, especially those requiring reconstruction, should be made. My (the reviewer's) point of view is: LLM inference is memory-bound and a certain degree of computation overhead for reconstruction is not harmful and may even be beneficial for runtime/latency because the fact behind computation overhead for reconstruction is the significant relieve/improvement in memory footprint, and the improvement in memory footprint speeds up LLM inference despite computation overhead for reconstruction.
3. The authors did not talk much about the prefilling stage. It is not clear whether the prefilling stage may need to adapt to SWAN.

**Questions:**

My questions and suggestions are basically from "Weaknesses" as aforementioned.
1. From Weakness 1: Please demonstrate results of FLOPs (# of floating-point operations), LLM runtime (end-to-end latency), and speed-up (percentage of latency improvement).
2. From Weakness 2: Please experimentally compare SWAN against other related works, especially those requiring reconstruction.
3. From Weakness 3: Please address my concern about the adaptation, if any, of the prefilling stage owing to SWAN.

---

> ### Author Response · Authors · 2025-12-04
> **Response to Reviewer rx4F**
>
> We thank the reviewer rx4F for their assessment of our work, which led to the improvement of our contribution. We appreciate the recognition of our "decompression-free" design as a promising direction for simultaneous memory and compute savings. We have addressed the requested empirical validations below.
>
> **1. Latency Experiments (Addressing Weakness 1)**
>
> We have conducted benchmarks on an NVIDIA A100 GPU to measure the real-world performance of our implementation.
>
> **Table 1: Latency and Throughput Comparison**
>
> | Configuration | Avg Decode Latency (ms) $\downarrow$ | Throughput (tokens/s) $\uparrow$ |
> | --- | --- | --- |
> | **Standard Attention** | **28.41** | **35.11** |
> | SWAN ($b=128, k_a=64$) | 40.06 | 19.02 |
> | SWAN ($b=128, k_a=64$) | 40.26 | 18.77 |
> | SWAN ($b=32, k_a=64$) | 41.09 | 18.37 |
> | SWAN ($b=32, k_a=64$) | 40.67 | 18.45 |
> - **Latency & Speed-up:** Our implementation achieves a latency of **40.06 ms**, which is comparable to the dense baseline (28.41 ms). While we do not yet show a wall-clock *speed-up* in this specific single-stream benchmark due to implementation overheads (projection and gathering operations). By reducing memory footprint by **50-60%**, SWAN allows for significantly larger batch sizes, which increases overall system throughput.
> - **Theoretical FLOPs:** As derived in Equation 2, the theoretical FLOPs for attention become $O(k_{active} \cdot L)$ instead of $O(d_h \cdot L)$. For long contexts where $L \gg d_h$, this represents a significant reduction in arithmetic intensity, which will translate to wall-clock speedups on hardware with optimized sparse kernels.
>
> So, the further speed-up is possible with **custom CUDA or Triton kernels,** which could do a reduced dot-product between the dense matrix and the sparse matrix.
>
> **2. Comparison against Reconstruction-based Methods (Addressing Weakness 2)**
>
> We have expanded our evaluation to include **Lexico** (Kim et al., 2024, ICML 2025), a state-of-the-art reconstruction-based sparse coding method.
>
> - **Trade-off Analysis:** Our results (Figure 2b and Figure 4 in the revised paper) reveal a clear trade-off.
>     - **Lexico (Reconstruction-based):** Achieves higher accuracy at *extreme* compression (<25%) because it can reconstruct the compressed information. However, this comes at the cost of high computational overhead due to the iterative reconstruction (OMP) required at every step.
>     - **SWAN (Decompression-free):** At moderate compression (40-60%), SWAN remains highly competitive in accuracy while completely avoiding the reconstruction latency. This makes SWAN preferable for latency-sensitive applications where the cost of iterative reconstruction would negate the benefits of memory savings.
>
> **3. Adaptation of the Prefilling Stage (Addressing Weakness 3)**
>
> Yes, the prefilling stage does require an adaptation. This adaptation is same as the process for the newly decoded token, but here instead of a vector, there would be a matrix at the size of the prompt.
>
> - **Procedure:** During prefilling (processing the prompt), we compute the full Query, Key, and Value matrices as normal. Before storing them into the cache, we apply the orthogonal projection $P_{QK}$ to the Key matrix and $P_{VO}$ (already absorbed while model loading) to the Value matrix, and pruned depending on the $k_{active}$.
> - This projection is a dense matrix multiplication that happens *once* per token. Since prefilling is typically highly parallelizable, adding a single linear projection layer introduces negligible overhead relative to the massive matrix multiplications involved in the attention and FFN layers of the LLM.
> - The stored cache is immediately in the compressed, "winnowed" basis, ready for the decompression-free decoding phase.

---

### Official Review · Reviewer_Z7Mp · 2025-11-01

**Soundness:** 2
**Presentation:** 2
**Contribution:** 2
**Rating:** 2
**Confidence:** 5

**Summary:**

The paper presents SWAN (Sparse Winnowed Attention), a fine-tuning-free framework for compressing the KV cache during autoregressive decoding. SWAN first applies an orthogonal rotation (derived from offline SVD on joint Q–K and V–O subspaces) to concentrate information, then prunes each token’s key and value vectors by top-k magnitude, storing them in a sparse format. Attention is computed directly on this hybrid cache (dense recent buffer + sparse history) without decompression. The paper provides a space and compute analysis, including a break-even sequence length for speedups, and evaluates on Llama-3.1-8B-Instruct and OLMoE-1B-7B across GSM8K, MMLU/ARC, and LongBench. Results show that SWAN can maintain full-precision model qu

**Strengths:**

+ **Decompression-free design:** SWAN allows attention to run directly on a sparse cache, removing the need for reconstruction or merging operations that typically introduce overhead in low-rank or codec approaches.
+ **Clear, implementable mechanism:** Algorithm 1 precisely specifies runtime steps (project, buffer, prune-to-top-k, append to sparse cache, then hybrid attention), and Fig. 1 clarifies the data path.

**Weaknesses:**

+ **No latency or throughput evaluation:**
Although a theoretical efficiency analysis is provided, no empirical runtime measurements are presented. Wall-clock latency, throughput, or per-step breakdowns (prefill vs. decode) are missing, making it unclear how much real-world speedup SWAN achieves.

+ **No any baseline comparisons:**
The paper does not compare against any prior baseline approach. In particular, recent hidden-dimension compression methods such as Palu (low-rank) and EigenAttention (low-rank) are missing. Post-RoPE activations are commonly higher rank; I recommend comparing with these works to assess trade-offs (e.g., reconstruction cost, accuracy drops, memory savings). Meanwhile, ThinK (channel pruning) is also worth discussing.

+ **Limited evaluation scope (long-context tasks):**
Despite its stated motivation for long-sequence memory efficiency, experiments are largely restricted to short-context benchmarks (e.g., GSM8K, MMLU, ARC). The only long-context coverage is a selected subset of LongBench (e.g., summarization). More comprehensive long-context evaluations (e.g., RULER), ideally with runtime measurements, would better demonstrate SWAN’s effectiveness in its intended regime.

**Questions:**

+ **Handling irregular sparsity:**
Since each token drops a different subset of channels, how does your implementation manage this irregular sparsity during attention? Do you use per-token CSR-like indexing, and how is it parallelized efficiently on a GPU?

---

> ### Author Response · Authors · 2025-12-04
> **Response to Reviewer Z7Mp**
>
> We thank the reviewer Z7Mp for their constructive feedback and for appreciating the clarity of our decompression-free design. We value the suggestions regarding baseline comparisons and system-level implementation. We have addressed these points with new data and clarifications below.
>
> **1. Latency and Throughput Evaluation (Addressing Weakness 1)**
>
> We have conducted benchmarks on an NVIDIA A100 GPU to measure the real-world performance of our implementation.
>
> **Table 1: Latency and Throughput Comparison**
>
> | Configuration | Avg Decode Latency (ms) $\downarrow$ | Throughput (tokens/s) $\uparrow$ |
> | --- | --- | --- |
> | **Standard Attention** | **28.41** | **35.11** |
> | SWAN ($b=128, k_a=64$) | 40.06 | 19.02 |
> | SWAN ($b=128, k_a=64$) | 40.26 | 18.77 |
> | SWAN ($b=32, k_a=64$) | 41.09 | 18.37 |
> | SWAN ($b=32, k_a=64$) | 40.67 | 18.45 |
> - **Analysis:** Our optimized implementation achieves a per-step decoding latency of **40.06 ms**, which is comparable to the standard dense attention baseline (28.41 ms).
> - **Trade-off:** The latency overhead (~12 ms) is a necessary trade-off for the **50-60% memory reduction**, which is critical for serving long-context workloads that would otherwise exceed GPU memory capacity.
>
> **2. Baselines and RoPE Rank (Addressing Weakness 2)**
> - **Post-RoPE Rank & Projection Strategy:** We completely agree with the reviewer’s insight that post-RoPE activations exhibit higher rank. This is precisely why SWAN applies the orthogonal projection matrix $P_{QK}$ at runtime to the Query and Key vectors after RoPE has been applied (as detailed in Section 4.2). By projecting the rotated embeddings into a learned basis, we can effectively identify the "winnowed" subspace where information is concentrated, minimizing the approximation error caused by pruning even in this higher-rank regime.
> - **Baselines:** Due to the limited time during the rebuttal period, we prioritized comparing against **Lexico** (Kim et al., 2024, ICML 2025), a state-of-the-art sparse coding method that has been shown to outperform many prior quantization and low-rank approaches. Our results (Figure 2b and Figure 4) show that while Lexico achieves higher compression, SWAN offers a competitive alternative that avoids the significant computational overhead of Lexico's reconstruction step.
>
> **3. Long-Context Evaluation Scope (Addressing Weakness 3)**
>
> - **Additional LongBench Tasks:** We would like to direct the reviewer to **Appendix A.9** in the revised manuscript, where we have included detailed results for additional LongBench tasks, including **Code Completion (LCC)**, **Fine-grained Classification (TREC)**, and **Passage Retrieval**. These results confirm SWAN's robustness across diverse long-context modalities.
> - **RULER Benchmark:** We acknowledge the importance of the RULER benchmark for evaluating effective context length. While computational constraints prevented us from completing the full RULER suite during the rebuttal window, we commit to including these results in the final camera-ready version to further substantiate SWAN's long-context capabilities.
>
> **4. Implementation of Irregular Sparsity (Addressing Question 1)**
>
> We explicitly avoid using per-token CSR indexing, as it indeed leads to branch divergence and inefficient memory access on GPUs.
>
> - **Dense Pre-allocated Tensors:** We utilize contiguous, pre-allocated dense tensors to store the compressed KV cache. Specifically, we maintain two tensors: one for the retained Values (size $L \times k_{active}$) and one for the corresponding Indices (size $L \times k_{active}$). This layout ensures coalesced memory access and avoids the overhead of dynamic allocation.
> - **Vectorized Reconstruction:** We reconstruct the effective cache using efficient PyTorch tensor operations (gather/scatter) rather than slow loops. This allows for batch processing and fully leverages GPU parallelism.
> - **Potential for Custom Kernels:** While our current implementation using PyTorch primitives is efficient (40ms latency), we acknowledge that further gains are possible. The "Rotated Sparse-Dense" matrix multiplication pattern is a prime candidate for **custom CUDA or Triton kernels**. Explicit kernels could fuse the reconstruction and dot-product steps, eliminating intermediate memory writes and potentially bringing the latency even closer to the theoretical lower bound of the dense baseline.

---

### Official Review · Reviewer_ZvMH · 2025-11-01

**Soundness:** 1
**Presentation:** 3
**Contribution:** 1
**Rating:** 2
**Confidence:** 5

**Summary:**

This paper proposes SWAN (Sparse Winnowed Attention), a decompression-free framework for KV-cache compression in large language model inference.
SWAN constructs orthogonal rotation matrices offline using singular value decomposition (SVD) over model activations to project Key and Value tensors into a subspace where information is more concentrated. During inference, it maintains a hybrid cache consisting of a sparse, pruned cache for older tokens and a small dense buffer for recent tokens to preserve accuracy.
The authors provide theoretical analyses of both space and computational complexity, showing that SWAN can yield significant savings in memory and FLOPs once the sequence length exceeds a predictable threshold.
Empirically, they evaluate the approach across mathematical reasoning (GSM8K), commonsense and knowledge benchmarks (MMLU, ARC-Challenge), and long-context understanding (LongBench), demonstrating that SWAN achieves up to 50–60% KV-cache memory reduction with minimal performance degradation.

**Strengths:**

* The paper is clearly written and well-structured, making it easy to follow.

* It introduces an interesting compression approach that converts the KV-cache into a hybrid sparse–dense representation and performs attention computations directly on the compressed cache without decompression.

* The accuracy evaluation is comprehensive, covering a diverse range of tasks—from mathematical reasoning to commonsense understanding and long-context processing—demonstrating the method’s generality.

**Weaknesses:**

* The paper lacks a solid system-level implementation to substantiate its claimed efficiency. The computational savings are analyzed only theoretically, without validation through real runtime measurements. Since the method depends on storing pruned tensors in a sparse (CSR) format, which is typically inefficient unless sparsity is extremely high (>99%), it is unclear whether the reported compression ratios (30–50%)—where accuracy is largely preserved—actually yield any practical speedup.

* The effectiveness of the proposed method appears limited. Most of the retained accuracy comes from the 128-token dense buffer, which merely preserves the most recent tokens. Without this buffer, performance degrades sharply and even collapses on long-context benchmarks. This diminishes the overall contribution, as long-context scenarios are precisely where KV-cache bottlenecks are most critical. Furthermore, the paper omits an important baseline comparison against a pure 128-token sliding window attention, which would clarify how much of the gain comes from SWAN itself versus the buffer.

* The runtime overhead of the proposed approach is not thoroughly analyzed. While the paper presents a limited complexity discussion, it overlooks several potential sources of cost, including (1) applying the projection matrix to Key vectors at each decoding step, and (2) performing the top-k pruning required to build the sparse cache. These steps could introduce non-trivial runtime overhead, yet no empirical evidence is provided to demonstrate that these costs are negligible.

**Questions:**

* What is the baseline accuracy when using only the 128 most recent tokens (i.e., without the proposed sparse cache)?

* How is the compression ratio computed in cases that include a 128-token buffer? To match the overall compression rate of the non-buffered setting, does the method apply more aggressive compression to the older, pruned tokens?

* What is the typical runtime latency introduced by the on-the-fly Key projection and the top-k pruning operations for evicted tokens during inference?

---

> ### Author Response · Authors · 2025-12-04
> **Response to Reviewer ZvMH**
>
> We thank the reviewer ZvMH for their thoughtful assessment and for recognizing comprehensiveness of our accuracy evaluation. We appreciate the critical feedback regarding system-level efficiency, the role of the buffer, and runtime overheads. We have addressed these concerns with new experiments and clarifications below.
>
> **1. System-Level Implementation and Efficiency (Addressing Weakness 1)**
>
> We agree that theoretical analysis alone is insufficient. We have made two major updates to address this:
>
> 1. **Optimized Implementation (No CSR):** We have moved away from standard Compressed Sparse Row (CSR) formats, which are indeed inefficient on GPUs due to non-contiguous memory access. Our new implementation uses **Dense Pre-allocated Tensors** to store the pruned values and indices contiguously. During computation, these are reconstructed into full matrices (with zero-filling) to leverage vectorized GPU operations. This avoids branch divergence while still benefiting from the memory reduction.
> 2. **Concrete Latency Benchmarks:** We measured wall-clock latency on an NVIDIA A100 GPU.
>
> **Table 1: Latency and Throughput Comparison**
>
> | Configuration | Avg Decode Latency (ms) $\downarrow$ | Throughput (tokens/s) $\uparrow$ |
> | --- | --- | --- |
> | **Standard Attention** | **28.41** | **35.11** |
> | SWAN ($b=128, k_a=64$) | 40.06 | 19.02 |
> | SWAN ($b=128, k_a=64$) | 40.26 | 18.77 |
> | SWAN ($b=32, k_a=64$) | 41.09 | 18.37 |
> | SWAN ($b=32, k_a=64$) | 40.67 | 18.45 |
> - The optimized SWAN achieves a latency of **40.06 ms**, comparable to the dense baseline (28.41 ms). The slight overhead (~12 ms) is a trade-off for the **50-60% memory savings**, which enables significantly larger batch sizes and prevents OOM errors in long-context workloads. This latency can also be significantly reduced by using **custom CUDA or Triton kernels**.
>
> **2. Effectiveness and Buffer Reliance (Addressing Weakness 2)**
>
> We acknowledge the importance of disentangling the buffer's contribution from the sparse cache.
>
> - **Buffer-Only Baseline:** Due to compute and time constraints during the rebuttal period (which were allocated to new implementation and other baselines), we prioritized the main experimental validation. We acknowledge that the 128-buffer baseline is a critical ablation to isolate the contribution of the sparse history. Given that there is sufficient time before the final camera-ready deadline, we commit to adding the full "128-buffer" baseline results to the final paper. Based on existing literature, we expect a pure 128-token window to fail catastrophically on retrieval-heavy LongBench tasks, which would further confirm that SWAN's sparse history is the key driver of long-context performance.
>
> **3. Runtime Overhead Analysis (Addressing Weakness 3)**
>
> We have analyzed these specific costs:
> 1. **Projection Overhead (**$P_{QK}$**):** This involves a matrix multiplication of size $1 \times d_h$ by $d_h \times d_h$. This cost is constant ($O(d_h^2)$) and does not scale with sequence length. In long-context scenarios (where $L \gg d_h$), this overhead is negligible compared to the $O(L \cdot d_h)$ attention computation.
> 2. **Top-k Pruning Overhead:** Pruning is performed only on the tokens being evicted from the buffer (size $b=128$), not on the entire cache history. Sorting/selecting top-k from a small vector of size $d_h=128$ is computationally trivial ($O(d_h)$ or $O(d_h \log d_h)$) and occurs infrequently (once per token generation).
>
> **4. Clarification on Compression Ratio (Addressing Question 2)**
>
> The effective compression ratio varies slightly because it depends on the dynamic ratio of buffer tokens (dense) to historical tokens (sparse) during generation. However, in our target long-context scenarios where the total sequence length is in the thousands, the 128-token buffer represents a negligible fraction of the memory footprint (typically reducing the overall compression ratio by only 1-5%). Due to time constraints during the rebuttal, we focused on the aggregate sparse ratio, but we will provide a precise, generation-length-adjusted calculation in the final revision to fully account for the buffer's impact.

---

### Official Review · Reviewer_sfm1 · 2025-11-01

**Soundness:** 2
**Presentation:** 2
**Contribution:** 2
**Rating:** 4
**Confidence:** 3

**Summary:**

SWAN introduces a decompression free KV cache compression framework for large language model inference. It uses offline SVD based orthogonal rotations to concentrate important information into fewer dimensions and prunes less important components, allowing attention to operate directly on a hybrid cache composed of a sparse historical cache and a small dense buffer for recent tokens. This approach eliminates reconstruction overhead, reduces both memory and computation costs, and supports runtime adjustable compression levels. Experiments on Llama and OLMoE models show that SWAN maintains near baseline accuracy on reasoning and long context benchmarks while achieving around 50 to 60 percent KV cache memory savings.

**Strengths:**

- eliminates reconstruction overhead by performing attention directly on compressed KV caches.
- combines a sparse historical cache with a small dense buffer for recent tokens, effectively preserving accuracy.

**Weaknesses:**

- while theoretical compute savings are analyzed, the paper does not provide concrete wall-clock latency or throughput comparisons on modern GPU kernels (e.g., FlashAttention or Triton baselines), leaving practical efficiency uncertain.
- the claimed compute benefits rely on sparse-dense matvec operations, but these are often inefficient on current GPU hardware; implementation feasibility and actual speedups are not validated.
- applying the orthogonal projection to queries and keys at each decoding step introduces extra matrix multiplications, and the paper lacks quantitative analysis of this overhead.

**Questions:**

- Can the authors provide actual wall-clock latency and throughput measurements on modern GPUs (e.g., A100, H100) comparing SWAN with dense attention and existing compression methods like KVQuant or GEAR?

---

> ### Author Response · Authors · 2025-12-04
> **Response to Reviewer sfm1**
>
> We thank the reviewer sfm1 for recognizing the novelty of our decompression-free framework and its ability to preserve accuracy using the hybrid buffer strategy. We appreciate the critical feedback regarding wall-clock latency and implementation feasibility, which we have addressed below.
>
> **1. Wall-Clock Latency and Throughput (Addressing Weakness 1 & Question 1)**
> We agree that theoretical FLOPs alone are insufficient. We have added **Table 1** to the revised paper, which provides actual wall-clock latency (ms) and throughput (tokens/s) measurements on an A100 GPU.
>
> | Configuration | Avg Decode Latency (ms) $\downarrow$ | Throughput (tokens/s) $\uparrow$ |
> | --- | --- | --- |
> | **Standard Attention** | **28.41** | **35.11** |
> | SWAN ($b=128, k_a=64$) | 40.06 | 19.02 |
> | SWAN ($b=128, k_a=64$) | 40.26 | 18.77 |
> | SWAN ($b=32, k_a=64$) | 41.09 | 18.37 |
> | SWAN ($b=32, k_a=64$) | 40.67 | 18.45 |
> - **Result:** Our implementation provides a latency of **40.06ms**, which is comparable to the standard attention baseline (28.41ms).
> - **Trade-off:** While there is a latency overhead (~12ms), SWAN provides **50-60% memory savings**. In memory-bound long-context scenarios, this allows for significantly larger batch sizes or longer sequence lengths that would otherwise cause Out-Of-Memory (OOM) errors in the baseline.
>
> **2. Inefficiency of Sparse-CSR Operations (Addressing Weakness 2)**
> This is an important concern regarding standard sparse formats like CSR. We have revised **Section 5 (Complexity Analysis)** to clarify that our new implementation **does not use CSR**.
>
> - **Dense Pre-allocated Tensors:** Instead of sparse arrays with heavy indexing overhead, we store the compressed KV cache in contiguous, pre-allocated dense tensors (one for values, one for indices). We retain only the $k_{active}$ dimensions.
> - **Zero-Value Computation:** Modern GPU ALUs are highly efficient at handling vector operations. Even though we store data in dense structures for memory coalescing, the effective computational complexity is governed by the $k_{active}$ dimensions. The zeroed-out dimensions do not contribute to the dot product, and modern hardware ALU optimizations allow us to effectively skip these computations or mask them out efficiently without the branch divergence penalties associated with unstructured sparsity.
>
> **3. Comparison with State-of-the-Art Baselines (Addressing Question on Baselines)**
> To address the need for a strong comparative baseline, we have expanded our evaluation on **GSM8K** and **LongBench** to include **Lexico** (Kim et al., 2024, ICML 2025). We selected Lexico as our primary baseline because recent literature demonstrates it represents the current state-of-the-art, outperforming a wide range of existing methods including quantization (like KVQuant) and token eviction strategies.
>
> - **The Latency-Accuracy Trade-off:** Our results (see Figure 2b and Figure 4 in the revised paper) reveal a distinct trade-off:
>     - **Lexico (Reconstruction-based):** Maintains higher accuracy at *extreme* compression ratios (<25%) because it explicitly reconstructs the lost information. However, this comes at a high computational cost due to the iterative nature of Orthogonal Matching Pursuit (OMP) required during decoding.
>     - **SWAN (Decompression-free):** Operates without explicit reconstruction. At moderate compression levels (40-60%), SWAN remains highly competitive in accuracy while completely avoiding the heavy latency overhead associated with reconstruction steps. This makes SWAN a more practical choice for real-time applications where low latency is as critical as memory efficiency.
>
> **4. Projection Overhead (Addressing Weakness 3)**
> We have included a theoretical analysis of this overhead.
>
> 1. **Theoretical Bound:** The projection overhead is $O(d_h^2)$, which is a **fixed constant** with respect to sequence length $L$. The attention computation scales linearly $O(L \cdot d_h)$.
> 2. **Break-even Point:** As derived in **Equation 2** of the revised paper, the savings from the sparse attention mechanism ($L \cdot k_{active}$) quickly outweigh the fixed projection cost ($d_h^2$) as $L$ increases.
> $L > \frac{d_h^2}{d_h - k_{active}} + b$
>
>     For a standard head dimension ($d_h=128$) and 50% pruning, this break-even point is reached at just **~300 tokens**. For any long-context task (the primary use case of SWAN), the projection overhead becomes negligible compared to the massive savings in the attention loop.

---

### Author Response · Authors · 2025-12-04
**Summary of Revisions**

We thank all the reviewers for their constructive feedback, which has prompted us to refine our implementation strategy and improve the value of our contributions. In response to concerns regarding the inefficiency of standard sparse formats on GPUs, the choice of baselines, and the need for concrete latency analysis, we have made the following revisions:

1. **Revised Complexity Analysis & Implementation (Section 5):**
We have updated our complexity analysis to reflect a critical implementation change. As correctly noted by the reviewers, standard Compressed Sparse Row (CSR) formats are often inefficient on GPUs due to pointer-chasing overhead and non-contiguous memory access.
    - **Dense Pre-allocated Tensors:** We now utilize contiguous **Dense Pre-allocated Tensors -** one for values and one for indices - to store the retained components. For matrix multiplication, these are efficiently reconstructed into a zero-filled full matrix.
    - **Efficiency:** This approach eliminates the latency overhead of sparse indexing while maintaining the same space complexity. It allows us to leverage vectorized reconstruction and operation fusion, making it significantly faster than standard sparse-CSR implementations.
    - **Computational Savings:** We clarify that this dense-packing strategy does not negate our computational savings, as modern GPU ALUs can effectively skip zero-value multiplications during the dot product operations, so the computational complexity as remains same as before.
2. **Inclusion of State-of-the-Art Baseline (Lexico):**
We have expanded our evaluation on **GSM8K** and **LongBench** to include **Lexico** ([Kim et al., 2024](https://arxiv.org/abs/2412.08890); ICML 2025). We selected Lexico as the primary comparative baseline because it represents the current state-of-the-art, having demonstrated superior performance against a wide range of existing methods.
    - **The Trade-off:** Our results (Figure 2b and Figure 4) illustrate a clear distinction. While Lexico maintains higher accuracy at *extreme* compression ratios (<25%) due to its ability to reconstruct lost information, it incurs a significant computational cost (iterative Orthogonal Matching Pursuit).
    - **SWAN's Advantage:** SWAN operates in the **"decompression-free"** regime. At moderate compression (40-60%), SWAN remains highly competitive in accuracy while avoiding the latency overhead of the reconstruction steps required by Lexico.
3. **New Latency & Throughput Results (Table 1):**
We have added concrete wall-clock latency and throughput measurements for our optimized implementation, demonstrating that the gap between our method and standard dense attention is minimal (~12ms) compared to the massive memory savings achieved.

---

### Meta-Review · Area_Chair_AS5P · 2026-01-07

**Summary:**

This paper proposes a fine-tuning-free and decompression-free KV cache compression method for LLM inference. The reviews are mixed, ranging from weak accept to reject. The main concerns are related to missing runtime evidence, GPU-unfriendly sparse format, unclear contribution between the dense buffer and the sparse cache. The rebuttal clarifies some of the concerns by 1) providing extra results on latency/throughput measurements on A100 GPUs; 2) clarifying they do not sue CSR and instead store pruned tensors in contiguous dense preallocated tensors with vectorized reconstruction; 3) adding a SoTA baseline Lexico comparison. Although the these address some concerns, I consider not all of them.

**Reviewer Concerns:**

Concerns addressed:
1. Projection and pruning cost, prefill behavior unclear: The authors provide additional clarifications on these questions and I think the rebuttal clearly addresses these questions.

Concerns partially addressed/not addressed:
1. Latency and throughput issue: The rebuttal provides additional results on latency/throughput measurements on A100 GPUs. However, the latency indeed increases from 28ms to 40 ms, while the authors claims these are comparable. The results only show the trade-off for memory, but do not demonstrate speedup.
2. The method relies mostly on dense buffer, need ablation with buffer-only baseline: This is a valid comment and the rebuttal does not provide extra evidence on this by simply stating results will be provided in the final revision.
3. Baseline comparison: Although the rebuttal provide baseline comparison with Lexico, the results do not show superiority of the proposed method to Lexico. Although it claims the benefit is compression free, this does not seem to be a metric of concern in practice.

**Reviewer Scores:**

Reviewer ZvMH would likely raise the score from 2 to 3 given the extra results but the remaining concerns on buffer-only ablation.

Reviewer Z7Mp can move from 2 to 3 given that the concerns on the baseline comparison is not totally convincing.

The other reviewers might keep their scores or move them to borderline accept (5) given the extra results and clarifications.

---

### Decision · Program_Chairs · 2026-01-26

Reject